# Assessment of potential transthyretin amyloid cardiomyopathy cases in the Brazilian public health system using a machine learning model

Isabella Zuppo Laper[1], Cecilia Camacho-Hubner[2], Rafaela Vansan Ferreira[1], Claudenice Leite Bertoli de Souza[3], Marcus Vinicius Simões[4], Fabio Fernandes[5], Edileide de Barros Correia[6], Ariane de Jesus Lopes de Abreu[1], Guilherme Silva Julian[3]*

1 IQVIA Brazil, São Paulo, São Paulo, Brazil, 2 Pfizer Global, New York City, New York, United States of America, 3 Pfizer Brazil, São Paulo, São Paulo, Brazil, 4 Medical School of Ribeirão Preto, University of São Paulo, São Paulo, Brazil, 5 Heart Institute, University of São Paulo, São Paulo, Brazil, 6 Instituto Dante Pazzanese de Cardiologia, São Paulo, Brazil

* guilherme.julian@pfizer.com

**Data Availability Statement:** All SIA-SUS and SIH-SUS data files belong to DATASUS and are

## Abstract

### Objectives

To identify and describe the profile of potential transthyretin cardiac amyloidosis (ATTR-CM) cases in the Brazilian public health system (SUS), using a predictive machine learning (ML) model.

### Methods

This was a retrospective descriptive database study that aimed to estimate the frequency of potential ATTR-CM cases in the Brazilian public health system using a supervised ML model, from January 2015 to December 2021. To build the model, a list of ICD-10 codes and procedures potentially related with ATTR-CM was created based on literature review and validated by experts.

### Results

From 2015 to 2021, the ML model classified 262 hereditary ATTR-CM (hATTR-CM) and 1,581 wild-type ATTR-CM (wtATTR-CM) potential cases. Overall, the median age of hATTR-CM and wtATTR-CM patients was 66.8 and 59.9 years, respectively. The ICD-10 codes most presented as hATTR-CM and wtATTR-CM were related to heart failure and arrythmias. Regarding the therapeutic itinerary, 13% and 5% of hATTR-CM and wtATTR-CM received treatment with tafamidis meglumine, respectively, while 0% and 29% of hATTR-CM and wtATTR-CM were referred to heart transplant.

available from the DATASUS website. DATASUS – Departamento de Informática do SUS. Portal da saúde - transferência de arquivos. Available from: http://www2.datasus.gov.br/DATASUS/index.php?area=0901.

**Funding:** This study was conducted by IQVIA Brazil and was sponsored by Pfizer Brazil. The funder provided support in the form of salaries for authors CCH, CLBS and GSJ but did not have any additional role in the study design, data collection and analysis, decision to publish, or preparation of the manuscript. The specific roles of these authors are articulated in the 'author contributions' section.

**Competing interests:** IZL, RVF and AJLA are full-time employees at IQVIA Brazil. CCH, CLBS and GSJ are full-time employees at Pfizer Brazil. This does not alter our adherence to PLOS ONE policies on sharing data and materials.

## Conclusion

Our findings may be useful to support the development of health guidelines and policies to improve diagnosis, treatment, and to cover unmet medical needs of patients with ATTR-CM in Brazil.

## Introduction

Amyloidosis are a group of protein misfolding disorders in which misfolded proteins form insoluble amyloid fibrils that deposit in the tissues leading to organ damage and dysfunction (1). Two types of amyloid account for 95% of cardiac amyloidosis: light-chain amyloid (AL) due to immunoglobin light-chain deposition and transthyretin (TTR) cardiac amyloidosis (ATTR-CM), which can be due to hereditary mutation (hATTR) or wild-type transthyretin (wtATTR) [1]. Hereditary transthyretin cardiac amyloidosis (hATTR-CM) is caused by one of the known heritable (autosomal dominant) mutations in the *TTR* gene, while wtATTR-CM (also known as senile or senile systemic amyloid CM) is caused by age-related changes in the wild-type TTR [1]. ATTR-CM is an under-recognized cause of heart failure (HF) in older adults, an important cardiovascular disease (CVD), which are still the leading cause of death worldwide. Although ATTR-CM is considered a rare cardiac disease, recent studies have shown a prevalence up to 13% of patients hospitalized with HF and preserved ejection fraction (HFpEF) [2]; 16% of patients with aortic stenosis undergoing transcatheter valve replacement [2]; 7–8% of patients undergoing carpal tunnel release surgery [3]; and 17% of older adults with HFpEF in an autopsy series [4]. Data on the epidemiology of the disease in Brazil are scarce, especially in the public health setting.

The natural history of ATTR-CM includes progressive HF, complicated by arrhythmias and conduction system disease. The clinical course is more variable for those with hATTR compared with wtATTR [1]. The hereditary form of the disease usually manifests itself after the age of 47, with a median survival ranging from 2 to 6 years after diagnosis (depending on genotype) for untreated patients [1, 5–7] due to its low penetrance. On the other hand, wtATTR-CM is a disease that predominately affects men >60 years of age, with a median survival ranging from 3.5 to 5 years after diagnosis in untreated patients (depending on the stage of the disease) [1, 5–7]. Diagnostic delays, which remain common in the current treatment landscape, are associated with particularly poor prognosis [8]. The diagnosis of ATTR-CM is challenging for several reasons, including the similarity of symptoms with HF, a prevalent and common disease, especially among older adults, and the unfamiliarity of clinicians with the disease and its appropriate diagnostic algorithm [1]. Misdiagnosis is common in ATTR-CM, contributing to diagnostic delays and risking both further disease progression and treatment with ineffective and potentially harmful therapies [9]. Management of cardiac amyloidosis is complex and specific for the type of amyloidosis that affects the patient. In Brazil, the only treatment approved for ATTR-CM is tafamidis, a TTR stabilizer that binds the thyroxine-binding sites of TTR with high affinity and selectivity, slowing dissociation of TTR tetramers into monomers, therefore inhibiting aggregation.

Given the misdiagnosis and significant morbidity of ATTR-CM and availability of treatment with TTR stabilization, it is essential to identify those ATTR-CM patients who are potentially under-recognized. Machine learning (ML) models applied to CVD [10–12] and based on medical claims data for the prediction of diseases and phenotypes have been described in the medical literature with increasing frequency [13–17]. In this way, this study aimed to identify and describe the profile of potential ATTR-CM patients treated in the Brazilian public health

system (SUS) and registered in inpatient and outpatient databases from DATASUS, using a predictive machine learning model.

## Methods

### Study design

This is a supervised machine learning study based on data analysis from a retrospective administrative outpatient and hospitalization databases that aimed to classify the database information and estimate the frequency of potential ATTR-CM cases in the Brazilian public health system model. The period of the analysis was from January 1, 2015, until December 30, 2021, in the database. The supervised model uses training datasets that contain information on the desired output (label; true outcome), i.e., the model learns from labelled training data how to predict the desired outcome. In this study, labelled data defined "reference ATTR-CM cases" and "not ATTR-CM" cases, based on criteria set by the investigators, validated by an expert panel. Then, the model classified and predicted the frequency among those cases labelled as "not ATTR-CM" which could be under-recognized ATTR-CM cases (ATTR-CM-like cases) (S1 Fig). The "reference" cases were those most likely to have a confirmed ATTR-CM diagnosis, while the "like" cases (which were the cases potentially classified as diagnosis related to ATTR-CM) were those most likely to be under-recognized ATTR-CM cases.

The ML approach was performed as follows:

- "ATTR-CM-reference" cases were defined as:

  - Reference wild-type ATTR-CM (wtATTR-CM): patients aged ≥ 50 years old, with at least one claim with wtATTR-related ICD-10 codes (Table 1), <u>AND</u> at least one claim with one of the following cardiac-related ICD-10 codes: I50.0, I50.1, I50.9, I11.0, I42.0, I42.1, I42.2, I35.0, I44.1, I44.2, I42.5. (Table 1).

  - Reference hATTR-CM: patients aged ≥ 18 years old, with at least one claim with hATTR-related ICD-10 codes (Table 1), **<u>AND</u>** at least one claim with one of the following cardiac-related ICD-10 codes: I50.0, I50.1, I50.9, I11.0, I42.0, I42.1, I42.2, I35.0, I44.1, I44.2, I42.5 (Table 1).

- Potential "ATTR-CM-like" cases were those not defined as ATTR-CM in the first step of the ML approach (*i.e.*, defined as "not ATTR-CM cases") but classified as ATTR-CM by the algorithm. For those patients, the criteria set for case classification were:

  - For wtATTR-CM:

> • Patients with at least one claim with any cardiac-related ICD-10 code (Table 1) **<u>AND</u>** at least one claim of the secondary procedures <u>AND</u> at least one mandatory procedure claim (S1 Table).

> • Patients with at least one claim with wtATTR-related ICD-10 codes (Table 1) **<u>AND</u>** at least one mandatory **<u>OR</u>** secondary procedure (S1 Table).

  - For hATTR-CM: patients with at least one claim with hATTR-related ICD-10 codes (Table 1) **<u>AND</u>** at least one mandatory **<u>OR</u>** one secondary procedure (S1 Table).

Once the potential "ATTR-CM-like" patients were filtered, the ML model evaluated these patients and classified it as "ATTR-CM like". The ICD-10 code related to amyloidosis was done based on previously published articles using claims data [18, 19].

## Data sources and feature selection

In Brazil, SUS is the universal healthcare system of which more than 75% (~ 150 million) of the population are exclusively dependent on. However, the remaining 25% of the population have supplementary private health insurance plans, and therefore may access SUS episodically [20]. This study was based on outpatient and inpatient administrative data from DATASUS, the Informatics Department of SUS, body responsible for collecting, processing, and disseminating healthcare data in Brazil [21]. Therefore, our study includes data from procedures performed in SUS perspective, which covers from 150 million until total population in Brazil (205 million inhabitants). Two datasets were considered: the Inpatient Information System (SIH [*Sistema de Informações Hospitalares*]) and Outpatient Information System (SIA [*Sistema de Informações Ambulatoriais*]). SIH and SIA are administrative databases for reimbursement purposes, not being able to analyze patient information related to medical charts level of details [22, 23]. The details of contents in the databases are described elsewhere [24] (S2 Fig).

Due to its administrative nature, SIH and SIA do not contain clinical data (e.g., signs and symptoms). Thus, cause of admission (as per International Classification of Diseases (ICD) 10 code) and procedures performed during the hospitalization were used as predictor variables. Additionally, data related to patient's age, state of residence, hospitalization, and outpatient visits date, diagnosis at entry (ICD based), procedures prescribed and performed, and in-hospital length of stay (days) were also extracted.

To build the model, a preliminary ICD-10 code list that could be potentially related to ATTR-CM (i.e., ICD-10 codes most presented or related to ATTR-CM) was created based on literature review. Subsequently, this list was validated by a group of ATTR-CM Brazilians experts, considering local caveats and codes most frequently used, and the final list with the ICD codes generated is presented at Table 1.

Different types of procedures were also selected based on the literature review [25, 26]. Codes and procedures reference names were collected from the Management System of Procedures, Medications and OPM of the Unified Health System (SIGTAP), which are the standard procedures approved within SUS [27]. The list of the selected procedures also validated by ATTR-CM experts considering local reality (S2 Table). The experts assessed each model step results to identify possible limitations (bias sources for labeling construction and confounding variables).

## Study population

Considering wtATTR-CM disease profile and its occurrence on patients that were 50 years [1] or older, as previously done in a similar study in the USA [25]. Therefore, for the wtATTR-CM cohort, were considered patients aged ≥ 50 years at index date (date of first procedure claim related to the ICDs selected as potentially associated with wtATTR-CM); with wtATTR-CM-related ICD-10 codes and any of the cardiac-related ICD-10 codes listed in Table 1 during the study period. And for the hATTR-CM cohort, were included patients aged ≥ 18 years at index date, with hATTR-CM-related ICD-10 codes during the study period.

Patients diagnosed with blood cancers, end-stage renal disease and cerebral amyloid angiopathy (ICD-10 codes C83.0, C83.3, C85.1, C90.0, C90.1, C88.8, C90.2, C90.3, C88.0, D47.2, D89.1, E88.0, N18.6 and I68.0) were excluded, minimizing the potential for our prediction model to overlap with other diseases and forms of cardiac amyloidosis. Also, as a quality step, patients with inconsistent data (e.g., negative age) or with ≥ 50% of missing data in the databases were excluded from analysis.

**Table 1.  Final International Classification of diseases (ICD-10) code list for identifying potential ATTR-CM cases in DATASUS.**

| Hereditary ATTR-CM | |
|---|---|
| E85.0 | Non-neuropathic heredofamilial amyloidosis |
| E85.1 | Neuropathic heredofamilial amyloidosis |
| E85.2 | Heredofamilial amyloidosis, unspecified |
| **Wild-type ATTR-CM** | |
| E85 | Amyloidosis |
| E85.3 | Secondary systemic amyloidosis |
| E85.4 | Organ-limited amyloidosis |
| E85.8 | Other amyloidosis |
| E85.9 | Amyloidosis, unspecified |
| **Cardiac-related** | |
| I50.0 | Congestive heart failure |
| I50.1 | Left ventricular failure, unspecified |
| I50.9 | Heart failure, unspecified |
| I51.7 | Cardiomegaly |
| I11.0 | Hypertensive heart disease with heart failure (congestive) |
| I35.0 | Aortic valve stenosis |
| I42.0 | Dilated cardiomyopathy |
| I42.1 | Hypertrophic obstructive cardiomyopathy |
| I42.2 | Other hypertrophic cardiomyopathies |
| I42.5 | Other restrictive cardiomyopathy |
| I44.0 | First degree atrioventricular block |
| I44.1 | Second degree atrioventricular block |
| I44.2 | Total atrioventricular block |
| I44.7 | Unspecified left bundle branch block |
| I47.2 | Ventricular tachycardia |
| I48.0 | Flutter and atrial fibrillation |
| Q25.3 | Aortic stenosis |
| G56.0 | Carpal tunnel syndrome |
| N18.0 | Chronic kidney disease |
| N18.8 | Other chronic kidney disease |
| N18.9 | Chronic kidney disease, unspecified |

This is a modelling study using secondary data from public information sources. In accordance with the study country regulation, studies using open access and anonymized databases do not require patient informed consent nor ethics committee approval [28].

## Linkage methods

Even though, Brazilian publicly available information from health information systems databases do not use a key standard identifier that allow observations on patient-level crossing different databases per legislation to guarantee data privacy, some outpatient databases have unique patient encrypted code (key identifier), which allows a probabilistic linkage approach with the hospitalization dataset.

Therefore, we performed a of the lack of patient match key, a probabilistic record linkage method to allow longitudinal assessment using SIH and SIA, following multiple steps with different combination of patient information from both databases, such as date of birth, city and

ZIP code [29]. Before each step, a data cleaning was performed to keep only good quality claims for linkage. About 5% of all patient records were discharged from analysis due to low quality information.

This approach, however, enables an assessment of each patient's longitudinal record and thus allows us to evaluate their journey across the system.

## Statistical methods

For the ML model approach, the selected data was separated in train (60% of patients), validation (20% of patients), and test (20% of patients) datasets (S3 Fig). During the training step, the model learned the patterns of the used data. The validation step, then, was used to decide the most suitable algorithm and the test was the final validation of the model. A supervised learning algorithm was fitted in the training set to learn the pattern of ATTR-CM and not ATTR-CM cases. We tested three different supervised algorithms (logistic regression, Support Vector Machine, XGBoost, and Random Forest), so we could choose the one with the best performance. A K-fold cross validation was performed to get the best model parameters and control overfitting. After evaluating the result of the best model in the validation set, it was also evaluated in the test set, to make sure this was the best model. The machine learning model approach is represented in S3 Fig. Data analysis was performed considering numerical and categorical variables. The continuous variables were described as measures of central tendency (mean, median) and spread, including the range, quartiles, absolute deviation, variance, and standard deviation, as applicable. The categorical variables were described as counts and percentages. The age variable was calculated based on the difference between the date of birth and the first ICD-10 code of interest reported (index date). The age was described as a continuous variable, including the mean, standard deviation, median and interquartile ranges; and by age groups (absolute number and proportion per category). The demographic variables were described as categorical variables, with absolute frequencies and percentage, as well as the frequency of the selected ATTR-CM-related ICD-10 codes. The proportion of ATTR-CM-reference and ATTR-CM-like cases among potential ATTR-CM cases in SUS was described by ATTR-CM type (hereditary and wild-type) per year. The model performance metrics were also evaluated, considering its accuracy, sensitivity, and specificity. The model performance metrics was also evaluated, considering its accuracy, sensitivity, and specificity, as follows: 1) Accuracy: the relationship between predicted vs. actual value, i.e., closeness of predicted value to the actual value; 2) Sensitivity: measured using predicted values of the output model with respect to changes in the input of the given model. It also computes the significance of attributes to obtain correct output; 3) Specificity: it is related to degree of confidence. The description of the equations used are included in S4 Fig.

Time of follow-up was calculated based on the difference between date of first claim of ICD-10 code of interest and the last date of patient information available at database. The annual hospitalization rate was described as the number of ATTR-CM-related hospitalizations per 100.000 inhabitants per each study year. The therapeutic itinerary was presented as the number and proportion of patients with record of tafamidis, heart transplant or liver transplant during the study period. The resource utilization per patient was summarized as the mean (SD) and median (IQR) number of hospital admissions and outpatient visits per each patient; and the resource utilization per patient per year (PPPY) was calculated as the median (95%CI) number of procedures divided by each patient's follow-up time in years, according to

the formula:

$$PPPY = {}^N visits/{}_F UP\ of\ each\ patient\ (in\ years)$$

## Results

### Study dataset construction and classification of the ATTR-CM cases

A total of 1,508,468 individuals with claims for the selected ICD-10 codes were identified in the database from 2015 to 2021. From those, 2,107 (0.14%) were excluded due to blood cancer, end-stage renal disease or cerebral amyloid angiopathy. Thus, 1,506,361 individuals were considered to start the construction of the hATTR-CM and wtATTR-CM cohorts (Fig 1).

Of the 1,506,361 individuals in the initial cohort, 860 were aged ≥ 18 years old and had at least one claim with hATTR ICD-10 codes, composing the hATTR-CM initial dataset. Of these, 477 hATTR-CM cases were identified from 2015 to 2021, of which 213 were classified as reference-hATTR-CM and 264 were classified as potential hATTR-CM cases in the first step of the ML model (Fig 1). Finally, among those cases classified as potential hATTR-CM cases in the first step of the algorithm, 49 (10.27%) were classified as hATTR-CM-like cases and 215 (45.07%) were classified by the ML model as non hATTR-CM cases (Table 2). That is, the model classified a total of 265 potential hATTR-CM patients (reference and like cases). The prevalence of hATTR-CM among hATTR patients was 24.8%, considering the 213 reference patients and the 860 individuals in the initial hATTR cohort.

Considering the construction of the wtATTR-CM cohort, 938,385 individuals were aged ≥ 50 years old and had at least one claim with wtATTR-CM ICD-10 codes or at least one

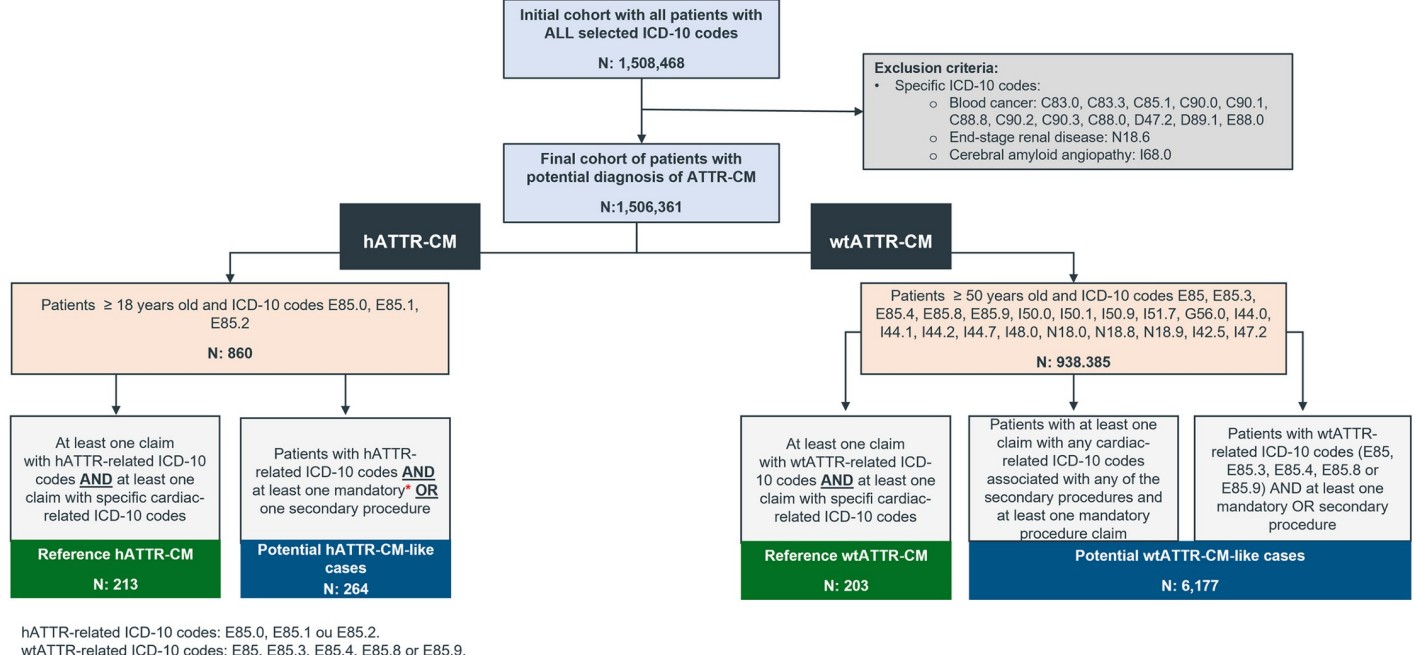

hATTR-related ICD-10 codes: E85.0, E85.1 ou E85.2.
wtATTR-related ICD-10 codes: E85, E85.3, E85.4, E85.8 or E85.9.
Specific cardiac-related ICD-10 codes: I50.0, I50.1, I50.9, I11.0, I42.0, I42.1, I42.2, I35.0, I44.1, I44.2, I42.5
General cardiac-related ICD-10 codes: I50.0, I50.1, I50.9, I51.7, G56.0, I44.0, I44.1, I44.2, I44.7, I48.0, N18.0, N18.8, N18.9, I42.5, I47.2

**Fig 1. Sample size construction flowchart.**

**Table 2. Proportion of ATTR-CM-reference and ATTR-CM-like cases in SUS according to the ML model classification from 2015 to 2021.**

|  | Total | 2015 | 2016 | 2017 | 2018 | 2019 | 2020 | 2021 |
|---|---|---|---|---|---|---|---|---|
| *Hereditary ATTR-CM* | **477** | **39** | **40** | **35** | **23** | **117** | **109** | **114** |
| Classification, N (%) |  |  |  |  |  |  |  |  |
| hATTR-CM-reference | 213 (44.65%) | 20 (51.28%) | 23 (57.5%) | 10 (28.57%) | 13 (56.52%) | 44 (37.61%) | 56 (51.38%) | 47 (41.23%) |
| hATTR-CM-like | 49 (10.27%) | 3 (7.69%) | 5 (12.5%) | 4 (11.43%) | 0 (0%) | 16 (13.68%) | 18 (16.51%) | 3 (2.63%) |
| *Non-hATTR-CM* | 215 (45.07%) | 16 (41.03%) | 12 (30%) | 21 (60%) | 10 (43.48%) | 57 (48.72%) | 35 (32.11%) | 64 (56.14%) |
| *Wild-type ATTR-CM* | **6380** | **2152** | **1117** | **999** | **774** | **593** | **342** | **403** |
| Classification, N (%) |  |  |  |  |  |  |  |  |
| wtATTR-CM-reference | 203 (3.18%) | 107 (4.97%) | 20 (1.79%) | 24 (2.4%) | 20 (2.58%) | 17 (2.87%) | 9 (2.63%) | 6 (1.49%) |
| wtATTR-CM-like | 1378 (21.6%) | 578 (26.86%) | 264 (23.63%) | 194 (19.42%) | 140 (18.09%) | 94 (15.85%) | 60 (17.54%) | 48 (11.91%) |
| *Non-wtATTR-CM* | 4799 (75.22%) | 1467 (68.17%) | 833 (74.57%) | 781 (78.18%) | 614 (79.33%) | 482 (81.28%) | 273 (79.82%) | 349 (86.6%) |

hATTR-CM: hereditary transthyretin amyloid cardiomyopathy; wtATTR-CM: wild-type transthyretin amyloid cardiomyopathy

claim with cardiac-related ICD-10 codes, being therefore included in the wtATTR-CM initial cohort. Of these, 203 were classified as reference-wtATTR-CM and 6,177 were classified as potential wtATTR-CM cases in the first step of the ML model (Fig 1). In the final step of the ML model, of the 6,177 cases classified as potential in the first step, 1,378 (21.6%) were classified as wtATTR-CM-like cases and 4,799 (75.22%) as non wtATTR-CM cases (Table 2). That is, the model classified a total of 1,581 potential wtATTR-CM patients (reference and like cases). The prevalence of wtATTR-CM cases was 21.6 cases per 100.000 patients ≥ 50 years old with cardiac-related ICD-10 codes, considering the 203 reference patients and the 938,385 individuals in the initial wtATTR-CM cohort. It is important to notice that this prevalence is based on an initial cohort of patients with cardiac failure and related diseases, and not on the overall Brazilian population.

## Machine learning model performance

The final validated model was applied to both hATTR-CM and wtATTR-CM datasets. The model classified the ATTR-CM cases as reference, potential or not ATTR-CM. For hATTR-CM cohort, the final validated model predicted 95.35% of hATTR-CM cases and 75.47% of not hATTR-CM cases and had an accuracy of 84.35% (Table 3). For the wtATTR-CM cohort (n = 6,380), the final validated model predicted 84.62% of wtATTR-CM cases and 77.85% of not wtATTR-CM cases and had an accuracy of 78.06% (Table 3).

## Demographic characteristics of ATTR-CM-reference and ATTR-CM-like patients

Overall, median age of hATTR-CM patients was 66.8 years (interquartile range [IQR] 50.5–70.3). In the reference group, median age was 66.8 (IQR 52.8–74.1) years, while in hATTR-CM-like group it was 65.9 (IQR 42.2–71.0). Most patients were over 60 years old over all groups, but a higher proportion of the age group from 30 to 49 years was observed in hATTR-CM-like. There were most males in general (58.8%), as in the reference hATTR-CM (58.2%) and hATTR-CM-like (61.2%) groups (Table 4).

The median age of wtATTR-CM patients was overall 59.9 years (IQR 55.1–66.3). In the wtATTR-CM reference group, median age was 65.9 (IQR 58.4–73.8) years, while in wtATTR-CM-like group it was 59.2 (IQR 54.8–65.2), demonstrating an opportunity to properly diagnose these potential patients while they're in a less advanced age. Most patients were

**Table 3. Model performance in final test.**

| *Hereditary ATTR-CM* | |
|---|---|
| Accuracy | 84.38% |
| Sensitivity | 95.35% |
| Specificity | 75.47% |
| *Wild-type ATTR-CM* | |
| Accuracy | 78.06% |
| Sensitivity | 84.62% |
| Specificity | 77.85% |

Accuracy: the proportion of correct classifications that a trained machine learning model achieves, i.e., the number of correct predictions divided by the total number of predictions across all classes; Sensitivity: measures the proportion of true positives thar are correctly identified by the model; Specificity: measures the proportion of true negatives that are correctly identified by the model.

under 70 years old over all groups, and the wtATTR-CM group had the higher proportion of individuals from 50 to 59 years old (49.1%). Males were the majority overall (62.1%), as well as in the hATTR-CM reference (58.6%) and hATTR-CM-like (62.6%) groups (Table 4).

Regarding the distribution of cases by state of residence, considering all patients (n = 262) and the hATTR-CM reference cohort (n = 213), most patients lived in São Paulo (31.5% and 26.5%, respectively), Minas Gerais (17.7% and 19.9%), Bahia (9.6% and 10.4%), Paraná (7.7% and 8.5%), and Rio de Janeiro (6.9% and 7.1%, respectively). For hATTR-CM-like cohort (n = 49), higher proportions of patients were observed in São Paulo (53.1%), Distrito Federal (10.2%), Minas Gerais (8.2%), Bahia (6.1%) and Rio de Janeiro (6.1%) (Table 5).

**Table 4. ATTR-CM patients demographic characteristics.**

| | hATTR-CM | | | wtATTR-CM | | |
|---|---|---|---|---|---|---|
| | **Total** | **hATTR-CM reference** | **hATTR-CM-like** | **Total** | **wtATTR-CM reference** | **wtATTR-CM-like** |
| | **262** | **213** | **49** | **1581** | **203** | **1378** |
| Age at index date | | | | | | |
| Mean ± SD | 61.6 ± 16.1 | 62.1 ± 15.9 | 59.4 ± 17.2 | 61.5 ± 8.3 | 66.7 ± 10.2 | 60.7 ± 7.6 |
| Median (min—max) | 66.8 (19.3–89.3) | 66.8 (19.3–89.3) | 65.9 (22.6–88.8) | 59.9 (50.0–102.1) | 65.9 (50.3–102.1) | 59.2 (50.0–91.8) |
| IQR | 50.5–73.0 | 52.8–74.1 | 42.2–71.0 | 55.1–66.3 | 58.4–73.8 | 54.8–65.2 |
| Age group, N (%) | | | | | | |
| 18 to 29 years | 10 (3.82%) | 8 (3.76%) | 2 (4.08%) | - | - | - |
| 30 to 39 years | 24 (9.16%) | 17 (7.98%) | 7 (14.29%) | - | - | - |
| 40 to 49 years | 30 (11.45%) | 23 (10.8%) | 7 (14.29%) | - | - | - |
| 50 to 59 years | 30 (11.45%) | 26 (12.21%) | 4 (8.16%) | 730 (46.17%) | 53 (26.11%) | 677 (49.13%) |
| 60 to 69 years | 65 (24.81%) | 52 (24.41%) | 13 (26.53%) | 570 (36.05%) | 71 (34.98%) | 499 (36.21%) |
| 70 to 79 years | 74 (28.24%) | 62 (29.11%) | 12 (24.49%) | 222 (14.04%) | 53 (26.11%) | 169 (12.26%) |
| ≥ 80 years | 29 (11.07%) | 25 (11.74%) | 4 (8.16%) | 59 (3.73%) | 26 (12.81%) | 33 (2.39%) |
| *Total available information* | *262* | *213* | *49* | *1581* | *203* | *1378* |
| Gender N (%) | | | | | | |
| Female | 108 (41.22%) | 89 (41.78%) | 19 (38.78%) | 600 (37.95%) | 84 (41.38%) | 516 (37.45%) |
| Male | 154 (58.78%) | 124 (58.22%) | 30 (61.22%) | 981 (62.05%) | 119 (58.62%) | 862 (62.55%) |

hATTR-CM: hereditary transthyretin amyloid cardiomyopathy; wtATTR-CM: wild-type transthyretin amyloid cardiomyopathy; IQR: interquartile range; SD: standard deviation

**Table 5. Distribution of ATTR-CM cases, from 2014 to 2021, according to the state of residence.**

| | hATTR-CM | | | wtATTR-CM | | |
|---|---|---|---|---|---|---|
| | Total | hATTR-CM reference | hATTR-CM-like | Total | wtATTR-CM reference | wtATTR-CM-like |
| | 262 | 213 | 49 | 1581 | 203 | 1378 |
| Acre | 0 (0%) | 0 (0%) | 0 (0%) | 1 (0.06%) | 1 (0.5%) | 0 (0%) |
| Alagoas | 1 (0.38%) | 1 (0.47%) | 0 (0%) | 14 (0.91%) | 3 (1.49%) | 11 (0.82%) |
| Amapá | 0 (0%) | 0 (0%) | 0 (0%) | 1 (0.06%) | 0 (0%) | 1 (0.07%) |
| Amazonas | 1 (0.38%) | 1 (0.47%) | 0 (0%) | 6 (0.39%) | 2 (1%) | 4 (0.3%) |
| Bahia | 25 (9.62%) | 22 (10.43%) | 3 (6.12%) | 125 (8.11%) | 12 (5.97%) | 113 (8.43%) |
| Ceará | 4 (1.54%) | 4 (1.9%) | 0 (0%) | 42 (2.73%) | 4 (1.99%) | 38 (2.84%) |
| Distrito Federal | 8 (3.08%) | 3 (1.42%) | 5 (10.2%) | 32 (2.08%) | 2 (1%) | 30 (2.24%) |
| Espírito Santo | 6 (2.31%) | 6 (2.84%) | 0 (0%) | 30 (1.95%) | 2 (1%) | 28 (2.09%) |
| Goiás | 16 (6.15%) | 14 (6.64%) | 2 (4.08%) | 32 (2.08%) | 11 (5.47%) | 21 (1.57%) |
| Maranhão | 0 (0%) | 0 (0%) | 0 (0%) | 0 (0%) | 0 (0%) | 0 (0%) |
| Mato Grosso | 1 (0.38%) | 1 (0.47%) | 0 (0%) | 7 (0.45%) | 0 (0%) | 7 (0.52%) |
| Mato Grosso do Sul | 5 (1.92%) | 3 (1.42%) | 2 (4.08%) | 9 (0.58%) | 2 (1%) | 7 (0.52%) |
| Minas Gerais | 46 (17.69%) | 42 (19.91%) | 4 (8.16%) | 166 (10.77%) | 32 (15.92%) | 134 (10%) |
| Pará | 1 (0.38%) | 1 (0.47%) | 0 (0%) | 1 (0.06%) | 1 (0.5%) | 0 (0%) |
| Paraíba | 3 (1.15%) | 3 (1.42%) | 0 (0%) | 14 (0.91%) | 1 (0.5%) | 13 (0.97%) |
| Paraná | 20 (7.69%) | 18 (8.53%) | 2 (4.08%) | 84 (5.45%) | 11 (5.47%) | 73 (5.45%) |
| Pernambuco | 4 (1.54%) | 3 (1.42%) | 1 (2.04%) | 111 (7.2%) | 11 (5.47%) | 100 (7.46%) |
| Piauí | 1 (0.38%) | 1 (0.47%) | 0 (0%) | 5 (0.32%) | 1 (0.5%) | 4 (0.3%) |
| Rio de Janeiro | 18 (6.92%) | 15 (7.11%) | 3 (6.12%) | 46 (2.99%) | 8 (3.98%) | 38 (2.84%) |
| Rio Grande do Norte | 3 (1.15%) | 3 (1.42%) | 0 (0%) | 8 (0.52%) | 1 (0.5%) | 7 (0.52%) |
| Rio Grande do Sul | 5 (1.92%) | 5 (2.37%) | 0 (0%) | 75 (4.87%) | 12 (5.97%) | 63 (4.7%) |
| Rondônia | 2 (0.77%) | 2 (0.95%) | 0 (0%) | 4 (0.26%) | 1 (0.5%) | 3 (0.22%) |
| Roraima | 0 (0%) | 0 (0%) | 0 (0%) | 1 (0.06%) | 0 (0%) | 1 (0.07%) |
| Santa Catarina | 5 (1.92%) | 4 (1.9%) | 1 (2.04%) | 51 (3.31%) | 6 (2.99%) | 45 (3.36%) |
| São Paulo | 82 (31.54%) | 56 (26.54%) | 26 (53.06%) | 620 (40.23%) | 77 (38.31%) | 543 (40.52%) |
| Sergipe | 3 (1.15%) | 3 (1.42%) | 0 (0%) | 53 (3.44%) | 0 (0%) | 53 (3.96%) |
| Tocantins | 0 (0%) | 0 (0%) | 0 (0%) | 3 (0.19%) | 0 (0%) | 3 (0.22%) |
| Unknown | 2 | 2 | 0 | 40 | 2 | 38 |

hATTR-CM: hereditary transthyretin amyloid cardiomyopathy; wtATTR-CM: wild-type transthyretin amyloid cardiomyopathy.

For wtATTR-CM cohort, considering all patients (n = 1,581), most patients lived in São Paulo (40.2%), Minas Gerais (10.8%), Bahia (8.1%), and Pernambuco (7.2%). The same trend was observed in wtATTR-CM-like group, with most patients living in São Paulo (40.5%), Minas Gerais (10.0%), Bahia (8.43%), and Pernambuco (7.46%). For wtATTR-CM reference cohort (n = 203), higher proportions of patients were observed in São Paulo (38.3%), Minas Gerais (15.9%), Bahia (6.0%), and Rio Grande do Sul (6.0%) (Table 5).

## Proportion of ICD-10 codes most presented as ATTR-CM-like cases in DATASUS

The ICD-10 codes most presented as hATTR-CM in the overall cohort (n = 262) were I50.0 Congestive Heart Failure (59.5%), I50.9 Heart Failure, unspecified (41.9%), N18.9 chronic kidney disease, unspecified (27.5%), I44.2 Total atrioventricular block (18.3%), and I48.0 Flutter and atrial fibrillation (16.4%) (Fig 2). Considering only the hATTR-CM reference cohort

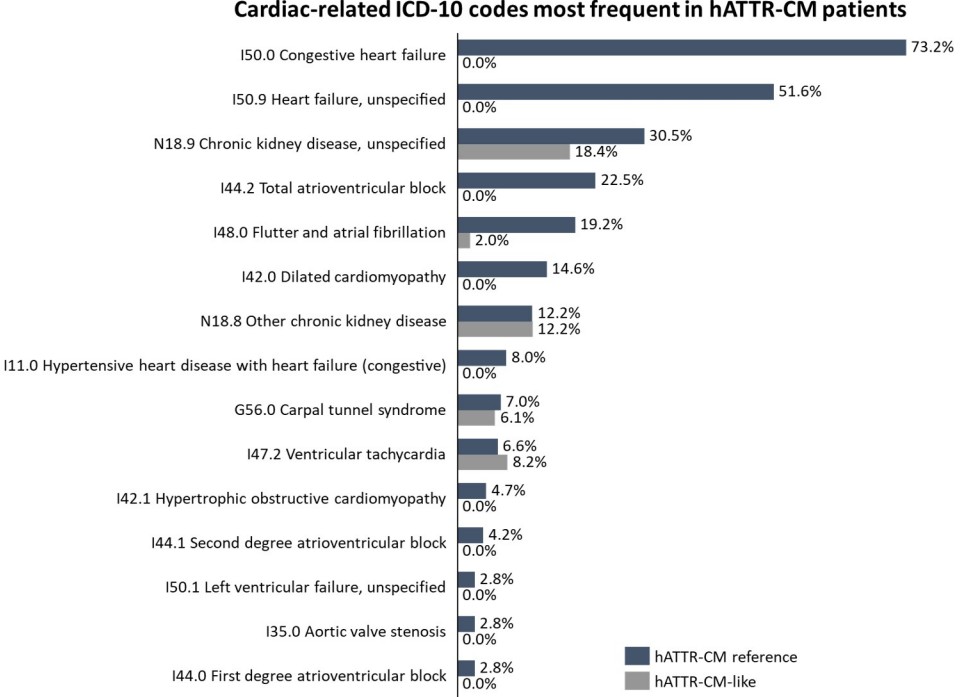

**Fig 2. Cardiac-related ICD-10 codes most frequent in hATTR-CM patients.**

(n = 213), most prevalent ICD-10 codes were I50.0 Congestive Heart Failure (73.2%), I50.9 Heart Failure, unspecified (51.6%), N18.9 chronic kidney disease, unspecified (30.5%), I44.2 Total atrioventricular block (22.5%), and I48.0 Flutter and atrial fibrillation (19.3%). For hATTR-CM-like cohort, on the other side, most common cardiac-related ICD-10 codes were N18.9 chronic kidney disease, unspecified (18.4%), N18.8 Other chronic kidney disease (12.2%), I47.2 Ventricular tachycardia (8.2%), and G56.0 Carpal tunnel syndrome (6.1%) (Fig 2).

For the wtATTR-CM cohort, heart failure and arrythmias were the ICD-10 codes most presented in the overall cohort (n = 1,581), with I50.0 Congestive Heart Failure (78.7%), I50.9 Heart Failure, unspecified (54.5%), I44.2 Total atrioventricular block (19.4%), I42.0 Dilated cardiomyopathy (18.4%), and N18.9 chronic kidney disease, unspecified (17.7%) (Fig 3). Considering only the wtATTR-CM reference cohort (n = 203), most prevalent ICD-10 codes were I50.0 Congestive Heart Failure (63.1%), I50.9 Heart Failure, unspecified (59.6%), I44.2 Total atrioventricular block (26.6%), and N18.9 chronic kidney disease, unspecified (26.6%%). For wtATTR-CM-like cohort (n = 1,378) most common cardiac-related ICD-10 codes were I50.0 Congestive heart failure (81.0%), I50.9 Heart failure, unspecified (53.8%), I42.0 Dilated cardiomyopathy (19.7%), I44.2 Total atrioventricular block (18.4%), and N18.9 chronic kidney disease, unspecified (16.3%) (Fig 3).

## Annual ATTR-CM hospitalization rate

Higher hospitalization rates were observed in the hATTR-CM reference compared to hATTR-CM-like group. The years with the higher hospitalization rates were 2017 and 2018, for hATTR-CM reference, and 2017, 2019 and 2020 for hATTR-CM-like cohort (Fig 4).

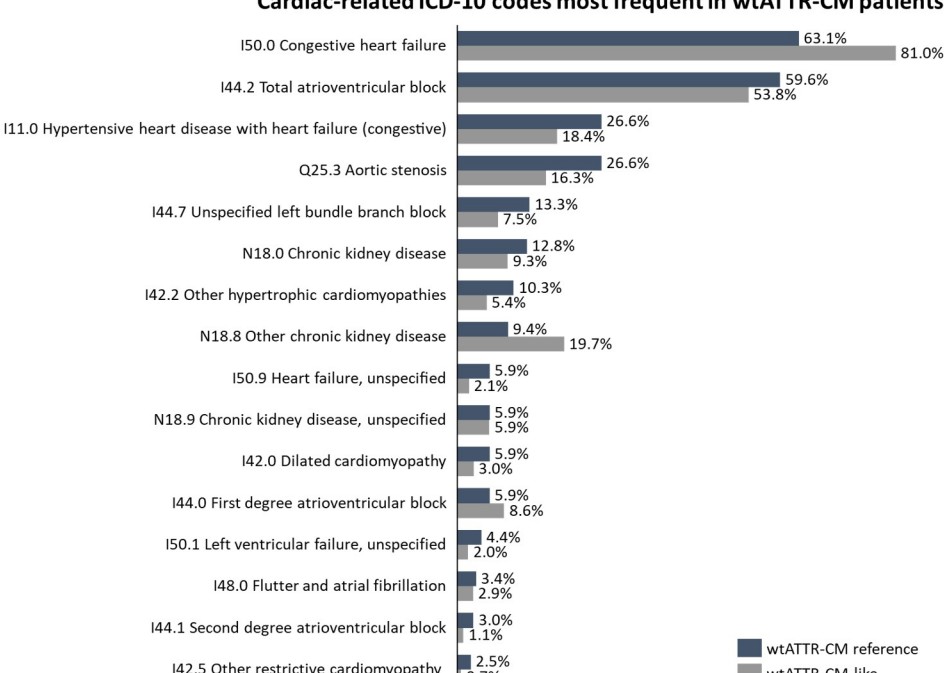

**Fig 3. Cardiac-related ICD-10 codes most frequent in wtATTR-CM patients.**

The opposite was observed for wtATTR-CM-like patients, which had a higher rate of hospitalization throughout the study period compared to wtATTR-CM reference patients. The years with the higher hospitalization rates were 2017 and 2018, for the wtATTR-CM reference, and 2015 and 2018 for wtATTR-CM-like cohort (Fig 5).

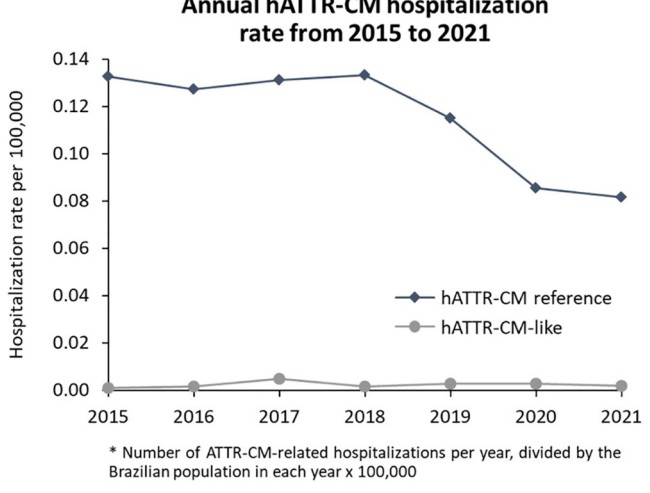

**Fig 4. Annual hATTR-CM hospitalization rate from 2015 to 2021.**

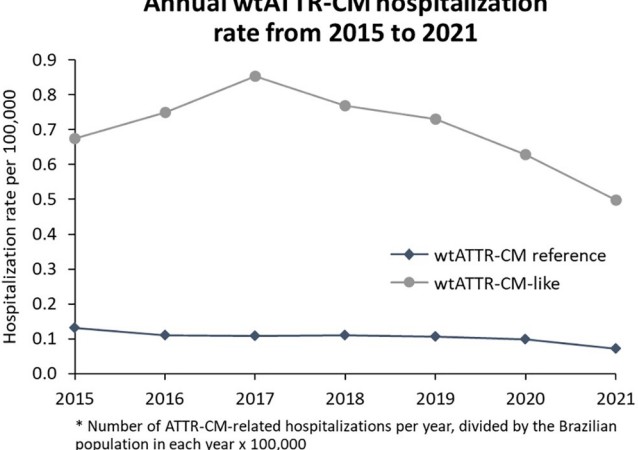

**Fig 5. Annual wtATTR-CM hospitalization rate from 2015 to 2021.**

## ATTR-CM therapeutic itinerary and healthcare resource utilization within SUS

**Inpatient setting.** Overall, hospitalizations were more frequent among hATTR-CM reference patients compared to hATTR-CM-like. For the entire cohort (n = 262), 217 (82.8%) patients had at least one record of hospitalization. The proportion of hospitalized patients decreased comparing hATTR-CM reference (94.4%) to hATTR-CM-like (32.7%). Median hospitalization rate per patient was 4.0 (IQR 2.0–8.0) for all patients, with similar trend in hATTR-CM reference (median [IQR 2.0–9.0]) and decreased in hATTR-CM-like (median 1.0 [IQR 1.0–2.2]). Median hospitalization rate per patient per year (PPPY) was 1.5 (IQR 1.0–2.2) for all patients. However, analysis by type of hATTR-CM revealed that hATTR-CM-like had a PPPY hospitalization rate 50% higher than hATTR-CM reference (1.50 [IQR 1.0–2.2] and 1.0 [1.0–1.6], respectively). The median number of days in hospital per patient (total days of hospitalization during follow up) was 6.0 (IQR 3.0–12.0) for all patients, with a decreasing trend comparing hATTR-CM reference (6.0 days [IQR 3.0–12.0]) to hATTR-CM-like (4.0 days [IQR 1.0–10.5] (Table 6).

**Outpatient setting.** In the outpatient setting, hATTR-CM-like patients had more outpatient visits compared to hATTR-CM reference. For the entire cohort (n = 262), 244 (93.1%) patients had at least one record of outpatient visit related to the hATTR. The proportion of patients with record of outpatient visits was higher in hATTR-CM-like (98.0%) compared to hATTR-CM reference (92.0%). Median number of outpatient visits per patient was 8.0 (IQR 2.8–20) for all patients. Median number of outpatient visits PPPY was 5.0 (IQR 2.0–8.0) for all patients. However, analysis by disease type revealed that hATTR-CM reference had less outpatient visits PPPY compared to hATTR-CM-like (5.0 [IQR 2.0–7.5] and 6.3 [IQR 1.0–8.7], respectively) (Table 6).

**Therapeutic itinerary.** For the entire cohort (n = 262), 35 (13.4%) patients had at least one claim for tafamidis meglumine, of which 31 (14,6%) were in the hATTR-CM reference group, and 4 (8.2%) were in the hATTR-CM-like group. There was no record of heart or liver transplant in the hATTR-CM cohort. The median number of treatment claims per hATTR-CM patient during the study period was 3.0 (IQR 1.5–11), varying from 3.0 (IQR 1.5–11) in the hATTR-CM reference cohort to 5.5 (IQR 1.8–9.8) in the hATTR-CM-like cohort.

**Table 6. Treatment patterns and healthcare resources utilization of ATTR-CM patients, from 2015 to 2021, per ATTR-CM type.**

| | hATTR-CM | | | wtATTR-CM | | |
|---|---|---|---|---|---|---|
| | **Total** | **hATTR-CM reference** | **hATTR-CM-like** | **Total** | **wtATTR-CM reference** | **wtATTR-CM-like** |
| | **262** | **213** | **49** | **1581** | **203** | **1378** |
| *Inpatient setting*[1] | | | | | | |
| Hospital admissions (n) | 1711 | 1677 | 34 | 11724 | 1532 | 10192 |
| Mean (SD) per patient | 7.9 (15.3) | 8.3 (15.8) | 2.1 (1.9) | 7.7 (13.5) | 7.8 (13.6) | 7.6 (13.5) |
| Median (IQR) per patient | 4 (2–8) | 4 (2–9.0) | 1 (1–2.2) | 4 (2–8) | 4 (2–7) | 4 (2–8) |
| Median (IQR) PPPY | 1.5 (1.0–2.2) | 1.5 (1.0–2.2) | 1.0 (1.0–1.6) | 1.8 (1.0–2.5) | 1.7 (1.0–2.2) | 1.8 (1.0–2.5) |
| Patients with at least one admission, n (%) | 217 (82.82%) | 201 (94.37%) | 16 (32.65%) | 1532 (96.9%) | 196 (96.55%) | 1336 (96.95%) |
| Total length of stay, days | | | | | | |
| Mean (SD) | 9.5 (11.4) | 9.5 (11.3) | 9.9 (16.2) | 10.5 (13.3) | 8.5 (10.8) | 10.7 (13.6) |
| Median (IQR) | 6.0 (3.0–12.0) | 6.0 (3.0–12.0) | 4.0 (1.0–10.5) | 6.0 (3.0–13.0) | 5.0 (2.0–10.0) | 7.0 (3.0–14.0) |
| *Outpatient setting*[1] | | | | | | |
| Outpatient visits (n) | 2752 | 2128 | 624 | 6321 | 886 | 5435 |
| Mean (SD) per patient | 11.3 (10.3) | 10.9 (10.1) | 13.0 (10.9) | 4.7 (12.1) | 5.0 (8.8) | 4.7 (12.5) |
| Median (IQR) per patient | 8 (2.8–20) | 7 (3–17.5) | 14 (2–24.2) | 2 (1–4) | 2 (1–4) | 2 (1–4) |
| Median (IQR) PPPY | 5.0 (2.0–8.0) | 5.0 (2.0–7.5) | 6.3 (1.0–8.7) | 1.0 (1.0–2.0) | 1.0 (1.0–2.0) | 1.0 (1.0–2.0) |
| Patients with at least one outpatient visit, n (%) | 259 | 196 | 63 | 1344 | 177 | 1167 |
| *Treatment metrics* | | | | | | |
| Treatment claims [2] | | | | | | |
| Mean (SD) per patient | 5.5 (4.6) | 5.5 (4.6) | 6.0 (5.4) | 6.4 (10.9) | 5.3 (7.4) | 6.5 (11.2) |
| Median (IQR) per patient | 3 (1.5–11) | 3.0 (1.5–11.0) | 5.5 (1.8–9.8) | 3 (2–6) | 3 (1–6) | 3 (2–6) |
| Median (IQR) PPPY | 2.0 (1.2–5.5) | 2.0 (1.2–5.5) | 4.0 (1.8–6.8) | 1.5 (1.0–2.1) | 1.4 (1.0–2.0) | 1.7 (1.0–2.2) |
| Patients with at least one claim for tafamidis, n (%) | 35 (13.36%) | 31 (14.55%) | 4 (8.16%) | 80 (5.06%) | 6 (2.96%) | 74 (5.37%) |
| Patients with at least one claim for heart transplant, n (%) | 0 (0%) | 0 (0%) | 0 (0%) | 457 (28.91%) | 1 (0.49%) | 456 (33.09%) |
| Patients with at least one claim for liver transplant, n (%) | 0 (0%) | 0 (0%) | 0 (0%) | 0 (0%) | 0 (0%) | 0 (0%) |
| Patients with claim for both heart and liver transplant, n (%) | (0%) | (0%) | (0%) | (0%) | (0%) | (0%) |
| *Procedure's metrics* | | | | | | |
| Procedures claims [3] | | | | | | |
| Mean (SD) per patient | 1 (0.0) | 1 (0.0) | 1 (0.0) | 2.0 (1.9) | 1.9 (1.9) | 2.0 (2.0) |
| Median (IQR) per patient | 1 (1–1) | 1 (1–1) | 1 (1–1) | 1 (1–2) | 1 (1–2) | 1 (1–2) |

(*Continued*)

**Table 6.** (Continued)

| | hATTR-CM | | | wtATTR-CM | | |
|---|---|---|---|---|---|---|
| | Total | hATTR-CM reference | hATTR-CM-like | Total | wtATTR-CM reference | wtATTR-CM-like |
| | **262** | **213** | **49** | **1581** | **203** | **1378** |
| Median (IQR) PPPY | 1 (1–1) | 1 (1–1) | 1 (1–1) | 1.0 (1.0–1.0) | 1.0 (1.0–1.0) | 1.0 (1.0–1.0) |

hATTR-CM: hereditary transthyretin amyloid cardiomyopathy; wtATTR-CM: wild-type transthyretin amyloid cardiomyopathy; IQR: interquartile range; PPPY: per patient per year; SD: standard deviation.

[1] Only inpatient or outpatient claims with the ICD-10 codes selected for the study (ATTR-CM-related or cardiac-related).

[2] Any treatment claim with the ICD-10 codes selected for the study (claims restricted to the selected ICD-10 codes). Treatment claims to be considered are: 06.04.54.006–0 tafamidis 20 mg, 05.05.02.004–1 Heart transplant, 05.05.02.005–0 Liver transplant (deceved donor), 05.05.02.006–8 Liver transplant (alive donor), 03.03.06.003–4 Treatment of hypertrophic heart disease, 03.03.06.023–9 Treatment of myocardiopathies, 03.03.06.002–6 Treatment of arrhythmias, 03.03.06.021–2 Heart failure treatment

[3] Any procedure claims with the ICD-10 codes selected for the study (claims restricted to the selected ICD-10 codes). Procedures claims to be considered are: 02.01.01.014–3 Heart biopsy, 02.08.05.002–7 Bone scintigraphy (full body), 02.08.05.001–9 Scintigraphy of joints and/or extremities and/or bone, 02.08.05.003–5 Bone scintigraphy with or without blood flow (full body), 02.07.02.001–9 Heart/aorta magnetic resonance, 04.03.02.012–3 Surgical treatment of carpal tunnel syndrome, 04.08.02.030–0 Tenosynovectomy in upper limb, 02.11.02.003–6 Electrocardiogram, 02.05.01.003–2 Transthoracic echocardiography, 02.05.01.002–4 Transesophageal echocardiography, 02.01.01.037–2 Skin and soft parts biopsy, 02.02.03.120–9 Troponin dosage, 02.02.03.128–4 Dosage BNP and NT-proBNP, 04.06.01.061–7 Endocavitary cardiac pacemaker implant, 04.06.01.062–5 Epimyocardial heart pacer implant, 04.06.01.063–3 Transvenous heart pacemaker implant, 04.06.01.064–1 Epimyocardial dual chamber pacemaker implant, 04.06.01.065–0 Transvenous dual chamber pacemaker implant, 04.06.01.066–8 Epimyocardial single chamber pacemaker implant, 04.06.01.067–6 Transvenous single chamber pacemaker implant, 04.06.01.069–2 Valve prosthesis implant

Median number of treatment claims PPPY was 2.0 (IQR 1.5–5.5) for all patients and for hATTR-CM reference cohort. The hATTR-CM-like group had a median number of treatment claims PPPY of 4.0 (IQR1.8–6.8) (Table 6).

**Procedure metrics.** The patterns of procedures claims were identical across disease types and for the entire cohort. Median number of procedures claims during the study period was 1.0 (IQR 1.0–1.0) and median PPPY was 1.0 (IQR 1.0–1.0). It was considered only claims of procedures defined for this study and related to the selected study ICD-10 codes (Table 6).

## Wild type ATTR-CM

**Inpatient setting.** Overall, hospitalizations were similar across wtATTR-CM reference and wtATTR-CM-like patients. For the entire cohort (n = 1,581), 1,532 (96.9%) patients had at least one record of hospitalization. The proportion of hospitalized patients was very similar between wtATTR-CM reference (96.6%) and wtATTR-CM-like (96.9%). Median hospitalization rate per patient was 4.0 (IQR 2.0–8.0) for all patients, with similar trend in wtATTR-CM reference and wtATTR-CM-like. Median hospitalization rate PPPY was 1.8 (IQR 1.0–2.5) for all patients. Analysis by type of wtATTR-CM revealed that wtATTR-CM-like and wtATTR-CM refence had a PPPY hospitalization rate almost identical (1.70 [IQR 1.0–2.2] and 1.8 [1.0–2.5], respectively). The median number of days in hospital per patient (total days of hospitalization during follow up) was 6.0 (IQR 3.0–12.0) for all patients, 5.0 (IQR 2.0–10.0) for wtATTR-CM reference and 7.0 (IQR 3.0–14.0) for wtATTR-CM-like (Table 6).

**Outpatient setting.** In the outpatient setting, outpatient visits seem similar across disease types. For the entire cohort (n = 1,581), 1,344 (85.0%) patients had at least one record of outpatient visit related to the wtATTR-CM. The proportion of patients with record of outpatient visits was higher in wtATTR-CM reference (87.2%) compared to wtATTR-CM-like (84.7%). Overall, the median number of outpatient visits per patient was 2.0 (IQR 1.0–4.0) for all

patients. Median number of outpatient visits PPPY was 1.0 (IQR 1.0–2.0) for all patients and across disease types (Table 6).

**Therapeutic itinerary.**    For the entire cohort (n = 1,518), 80 (5.1%) patients had at least one claim for tafamidis meglumine, of which 6 (3.0%) were in the wtATTR-CM reference group, and 74 (5.4%) were in the wtATTR-CM-like group. Overall, 28.9% of the wtATTR-CM cohort had record of heart transplant. Analysis by disease type revealed that from these, 1 (0.5%) was in the wtATTR-CM reference group and 456 (33.1%) were in the wtATTR-CM-like group. The median number of ATTR-CM-related treatment claims during the study period was 3.0 (IQR 2.0–6.0) for the entire cohort, with similar trends between wtATTR-CM reference and wtATTR-CM-like. Median number of treatment claims PPPY was 1.5 (IQR 1.0–2.1) for all patients, varying from 1.4 (IQR 1.0–2.0) in the wtATTR-CM reference cohort to 1.7 (IQR 1.0–2.2) in the hATTR-CM-like group (Table 6).

**Procedure metrics.**    The patterns of procedures claims were very similar across disease types and for the entire cohort. Median number of procedures claims during the study period was 1.0 (IQR 1.0–2.0) and median PPPY was 1.0 (IQR 1.0–1.0). Were considered only claims of procedures defined for this study and related to the selected study ICD-10 codes (Table 6).

## ATTR-CM-related procedures

Overall, most performed procedures in the hATTR-CM cohort were tafamidis meglumine (13.4%), specialized medical visits (5.0%), transthoracic echocardiogram (4.6%), and pre-transplant procedures (3.8%). For hATTR-CM reference cohort, most performed procedures were tafamidis meglumine (14.6%), pre-transplant procedures (4.2%), and transthoracic echo-cardiogram (3.8%). Considering specifically the hATTR-CM-like group, most performed pro-cedures were specialized medical visits (14.3%), transthoracic echocardiogram (8.2%), magnetic resonance of the heart (6.1%), cardiac catheterization (6.1%), and gabapentin (4.1%) (Fig 6 and S3 Table).

In the wtATTR-CM cohort, the most performed procedures were tafamidis meglumine (13.4%), specialized medical visits (5.0%), transthoracic echocardiogram (4.6%), and pre-transplant procedures (3.8%). For hATTR-CM reference cohort, most performed procedures were tafamidis meglumine (14.6%), pre-transplant procedures (4.2%), and transthoracic echo-cardiogram (3.8%). Considering specifically the hATTR-CM-like group, most performed pro-cedures were specialized medical visits (14.3%), transthoracic echocardiogram (8.2%), magnetic resonance of the heart (6.1%), cardiac catheterization (6.1%), and gabapentin (4.1%) (Fig 7 and S4 Table).

## Discussion

This study used a validated ML model to identify potential ATTR-CM cases in Brazilian National Health System (SUS). The results allowed to characterize demographically the ATTR-CM patients and to assess the proportion of ATTR-CM-reference and ATTR-CM-like cases among potential ATTR-CM cases. In addition, the study results showed the ICD-10 codes most presented as ATTR-CM-like cases in DATASUS, the annual hospitalization rate, the treatment patterns of ATTR-CM and ATTR-CM-like cases under SUS treatment and, finally, the average HCRU of ATTR-CM-reference and ATTR-CM-like cases. To the best of our knowledge, this is the first time that a ML model is used to assess potential ATTR-CM cases in the Brazilian national health system.

Overall, our final validated ML model had a good performance for classifying ATTR-CM cases from a retrospective analysis approach, in line with other predictive studies using ML models [13, 17, 25]. The accuracy was 78.06% for wtATTR-CM and 84.4% for hATTR-CM.

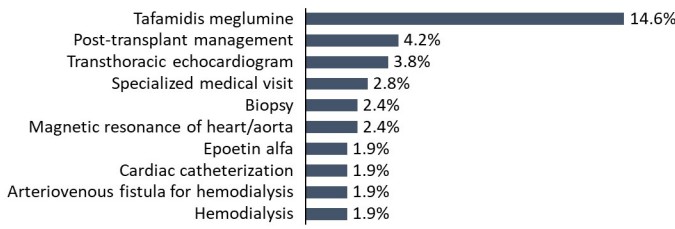

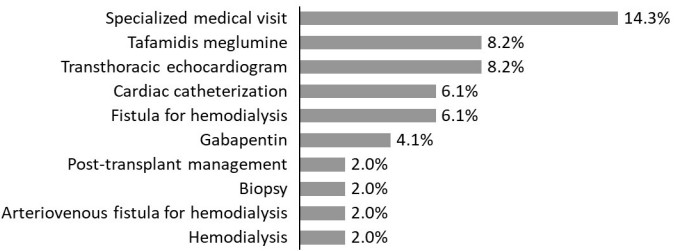

**Fig 6. Most frequent procedures in hATTR-CM reference and hATTR-CM-like cohorts from 2015 to 2021.**

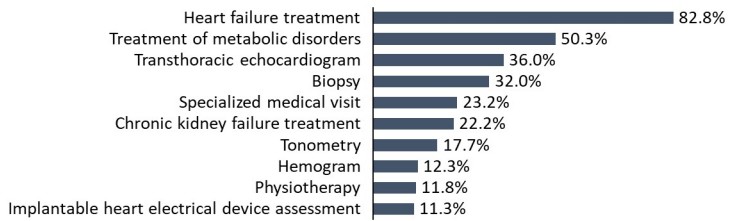

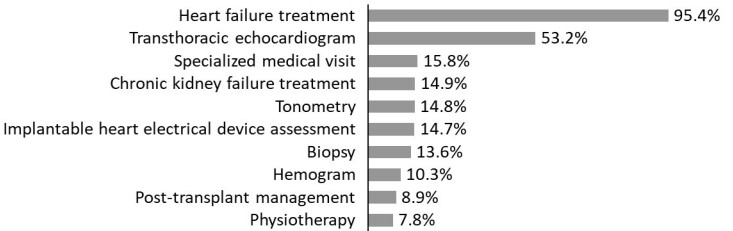

**Fig 7. Most frequent procedures in wtATTR-CM reference and wtATTR-CM-like cohorts from 2015 to 2021.**

We identified possible under-recognized ATTR-CM cases from the data available in DATASUS. According to the final classification of the ML model, 10.3% of hATTR-CM patients and 21.6% of wtATTR-CM patients (potential ATTR-CM cases) may have been under-recognized between 2015 and 2021. The delay in the diagnoses and the misdiagnoses of ATTR is often reported in literature [25, 30, 31]. The reasons for diagnostic delay are multifactorial and include symptom overlap with other conditions, low disease awareness, the historical need for invasive diagnosis, and until recently the lack of a disease-modifying treatment [30]. A previous study also identified under-recognized ATTR-CM cases in 4 different databases from USA, using ML model [25]. The authors highlighted the importance of ML model as a tool to help in the early diagnostic, resulting in a good prognostic of the disease [25].

A smaller proportion of under-recognized cases was identified in the hATTR-CM cohort compared to wtATTR-CM, probably because this cohort was built over a more restrict population, that is, patients with hereditary amyloidosis ICD-10 codes. On the other hand, in the wtATTR-CM cohort there was a higher proportion of underdiagnosed patients. While only 203 patients were classified as reference by the ML model, 1,581 patients were classified as wtATTR-CM-like cases, which may be indicative of a higher prevalence of the disease among older adults than expected. Previous studies have demonstrated that the clinical overlap between wtATTR-CM and other heart failure aetiologies is high [25, 30, 32], what can explain our finding.

Diagnosing ATTR-CM can be difficult, mainly for the wild-type, as cardiac symptoms are consistent with more common types of heart failure and the extra-cardiac manifestations are heterogeneous and nonspecific [30]. In one study developed in Spain, 30% of the patients with ATTR-CM had previously been misdiagnosed with other cardiac diseases such as: hypertensive heart disease, hypertrophic cardiomyopathy and ischemic heart disease [32]. In our study, the ICD-10 codes most related with ATTR-CM presented a similar profile to these findings.

Some patients with hATTR-CM may have a mixed phenotype, with cardiac and neurological manifestations. In our study, the prevalence of cardiac manifestations among hATTR patients was 24.8%, which is in line with the literature [33]. Previous studies have demonstrated that in patients with mixed phenotype the diagnostic delay is shorter than in patients with only cardiac manifestations [30, 34]. The early diagnosis can result in a better disease prognosis and in an adequate treatment, since a disease-modifying treatment is available for ATTR-CM treatment in Brazil.

Regarding the geographic distribution of the patients, a higher concentration was observed in the South and Southeast regions in this study, probably because these are the Brazilian regions with the higher number of specialized hospitals and clinics [35, 36]. Access to medical care in Brazil is widely influenced by the concentration of services in large urban centres [37]. The territorial extension of Brazil makes it even more important and challenging to provide a highly coordinated multi-layered healthcare system [37]. Therefore, it was expected a higher proportion of patients referred to more developed regions.

Concerning the therapeutic itinerary, tafamidis meglumine is the only specific drug treatment available in SUS for ATTR-CM treatment. In addition to tafamidis meglumine, heart transplantation is also available, however, due to its invasive characteristic it is considered only in extreme cases [15]. For the hATTR-CM patients, 13% received treatment with tafamidis meglumine and there was no record of heart transplant, while for wtATTR-CM patients only 5% were treated with tafamidis meglumine and 29% were referred to a heart transplantation. This data call attention, one more time, for the importance of an early diagnosis in the disease progression. wtATTR-CM patients, once early diagnosed, could have received the drug treatment, avoiding the heart transplantation [26, 31]. It is important to note that for hATTR-CM patients, although they were expected to have heart and liver transplantation [38, 39], it was

not identified in the study period. However, procedures related to the management of post-transplant patient and the use of tacrolimus, a drug used to prevent transplant organ rejection, appeared among the most common procedures i.e., these patients may have had transplantation before 2015 and were performing the maintenance during the study period. Another consideration refers to low number of heart and liver transplant in the study period and the introduction of tafamidis meglumine in SUS in 2016, which could be related, as previously demonstrated in a 20-year retrospective study of the Familial Amyloidosis Polyneuropathy World Transplant Registry [38].

The hospitalization rate and the resource utilization also evidence unmet medical needs, especially for wild-type patients; although a formal comparison was not performed, the hospitalization rate was much higher in the ATTR-CM-like group than in the ATTR-CM-reference group. These patients may have more hospitalizations records because lack of correct diagnosis and, consequently, lack of proper disease management. A previous study demonstrated that the use tafamidis meglumine was associated with a lower rate of hospitalization as well as a shorter length of stay per hospitalization among all treated patients, mainly when the treatment was initiated in patients at early disease stage [40].

Moreover, reductions in hospitalization rates might occur due to several factors [41]. In 2020 and 2021, however, these factors were compounded by the effect of the pandemic caused by the novel coronavirus [41]. In Brazil, the entire healthcare system was impacted, not only by the demand for care of COVID-19 cases, but also by the isolation and social distancing measures that compromised people's access to healthcare services [41]. In this study, we identified a reduction in hospitalization rates for both wtATTR-CM and hATTR-CM cohorts, probably related to the isolation and social distancing arising from COVID-19 pandemic.

Our study has some limitations. The use of retrospective data from administrative sources did not allow us to explore deeply and assess properly potential label biases and confounding variables, since information's like clinical data (e.g., signs and symptoms) were not available. To mitigate this, we used an expert panel assessment for each step of model labels construction, validation, classification performed, and frequency results obtained.

Additionally, there is an intrinsic limitation on the use of retrospective data is that the data, which are often incomplete, and this study depended on the quality and filling of non-mandatory data available [42].

Due to the administrative characteristics of the databases that were used, few clinical information was available, therefore the only specific predictive variables for identifying ATTR-CM-related cases in the model were ICD-10 code and procedures performed. Another important limitation is that DATASUS uses the International Classification of Diseases (10th version), which does not have a code for ATTR-CM, so we conducted the study based on the assumption of using a set of parameters validated by experts and key opinion leaders (including ICD-10 code and clinical procedures performed) for predicting ATTR-CM cases. The absence of laboratory test results available in the datasets were also a limitation for the label construction.

To reduce the probability of including patients with diseases different from ATTR-CM, we have defined, along with the expert pane, very specific mandatory procedures for classifying ATTR-CM and ATTR-CM-like cases. This might have an impact in the number of patients identified by the model for a few reasons.

Firstly, these are more expensive and specialized procedures. In the context of the Brazilian public health system, it is expected that there are barriers to access these type of procedures [43]. Therefore, it is supposed that there is a lower number of patients undergoing these procedures, which might have been reflected in the model results.

Secondly, we believe that the suspicion of amyloidosis in patients with heart failure is likely to be restricted to large centres (e.g., teaching hospitals, specialized clinics), as many physicians may not be familiar with ATTR-CM management [1], which can result in fewer patients who underwent diagnostic confirmation procedures.

Blind spots in machine learning can reflect the worst societal biases, with a risk of unintended or unknown accuracies in minority subgroups, and there are concerns over the potential for amplifying biases present in the data collected, which might lead to discriminatory bias [44]. In this context, the assessment of the expert panel in all model steps might have contributed to mitigating this influence.

## Conclusion

The outcomes found in this study supported the identification of potential ATTR-CM cases in DATASUS using a validated ML model, reflecting the public health system in Brazil. In our study, we were able to characterize this population demographically, clinically (considering their ICD-10 codes and procedures performed), and to identify the HCRU related to ATTR-CM management. The use of ML as a tool to identify potential patient of underdiagnosed diseases can be a hallmark for public health resources allocation and medical education strategies. In addition, our findings may be useful to support the development of health guidelines and policies to improve diagnosis, treatment and to cover unmet medical needs of patients with ATTR-CM in Brazil.

## Supporting information

**S1 Fig. Supervised machine learning model approach.**
(DOCX)

**S2 Fig. DATASUS databases description.**
(DOCX)

**S3 Fig. Machine learning model approach.**
(DOCX)

**S4 Fig. Equations for accuracy, sensitivity and specificity calculation.**
(DOCX)

**S1 Table. Mandatory and secondary procedures considered for classifying ATTR-CM and ATTR-CM-like cases.**
(DOCX)

**S2 Table. List of SUS standard procedures selected for the model.**
(DOCX)

**S3 Table. Top 20 procedures for hATTR-CM patients (number of unique patients).**
(DOCX)

**S4 Table. Top 20 procedures for wtATTR-CM patients (number of unique patients).**
(DOCX)

## Acknowledgments

The authors also thank Lucas Sozzi de Jesus and Lays Leonel for their medical writing support, and Ramon Pereira in the revision of the analysis.

## Author Contributions

**Conceptualization:** Isabella Zuppo Laper, Marcus Vinicius Simões, Ariane de Jesus Lopes de Abreu.

**Data curation:** Rafaela Vansan Ferreira.

**Formal analysis:** Rafaela Vansan Ferreira.

**Methodology:** Isabella Zuppo Laper, Rafaela Vansan Ferreira.

**Project administration:** Claudenice Leite Bertoli de Souza.

**Supervision:** Cecilia Camacho-Hubner, Claudenice Leite Bertoli de Souza, Fabio Fernandes, Edileide de Barros Correia, Ariane de Jesus Lopes de Abreu, Guilherme Silva Julian.

**Validation:** Isabella Zuppo Laper, Rafaela Vansan Ferreira, Marcus Vinicius Simões.

**Visualization:** Isabella Zuppo Laper.

**Writing – original draft:** Isabella Zuppo Laper.

**Writing – review & editing:** Cecilia Camacho-Hubner, Rafaela Vansan Ferreira, Claudenice Leite Bertoli de Souza, Marcus Vinicius Simões, Fabio Fernandes, Edileide de Barros Correia, Ariane de Jesus Lopes de Abreu, Guilherme Silva Julian.

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
