## [Decision Letter · Decision Letter 0]

25 Jul 2023

PONE-D-22-32110Assessment of potential transthyretin amyloid cardiomyopathy cases in the Brazilian public health system using a Machine Learning ModelPLOS ONE

Dear Dr. Silva Julian,

Thank you for submitting your manuscript to PLOS ONE. After careful consideration, we feel that it has merit but does not fully meet PLOS ONE’s publication criteria as it currently stands. Therefore, we invite you to submit a revised version of the manuscript that addresses the points raised during the review process.

We look forward to receiving your revised manuscript.

Kind regards,

Giuseppe Limongelli

Academic Editor

PLOS ONE

Journal Requirements:

"This study was funded by Pfizer Brazil"

"IZL, RVF and AJLA are full-time employees at IQVIA Brazil. CCH, CLBS and GSJ are full-time employees at Pfizer Brazil."

We note that one or more of the authors are employed by a commercial company: IQVIA Brazil, Pfizer Brazil 

(2) Please also provide an updated Competing Interests Statement declaring this commercial affiliation along with any other relevant declarations relating to employment, consultancy, patents, products in development, or marketed products, etc.  

Within your Competing Interests Statement, please confirm that this commercial affiliation does not alter your adherence to all PLOS ONE policies on sharing data and materials by including the following statement: ""This does not alter our adherence to  PLOS ONE policies on sharing data and materials.” (as detailed online in our guide for authors http://journals.plos.org/plosone/s/competing-interests) . If this adherence statement is not accurate and  there are restrictions on sharing of data and/or materials, please state these. 

Please note that we cannot proceed with consideration of your article until this information has been declared.

5. Please ensure that you refer to Figure 1 in your text as, if accepted, production will need this reference to link the reader to the figure.

Reviewers' comments:

Reviewer's Responses to Questions

**Comments to the Author**

1. Is the manuscript technically sound, and do the data support the conclusions?

Reviewer #1: Partly

Reviewer #2: No

2. Has the statistical analysis been performed appropriately and rigorously? 

Reviewer #1: No

Reviewer #2: Yes

3. Have the authors made all data underlying the findings in their manuscript fully available?

Reviewer #1: Yes

Reviewer #2: Yes

4. Is the manuscript presented in an intelligible fashion and written in standard English?

Reviewer #1: Yes

Reviewer #2: Yes

5. Review Comments to the Author

Reviewer #1: The authors use machine learning models to identify and classify ATTR-CM, an amyloid fibril that can cause cardiac arrest by preventing the ventricles from pumping blood effectively to the rest of the body. The study was conducted using patient data collected between 2015 and 2021 in order to assist the Brazilian health care system in enhancing patient health guidelines.

-In the introduction I suggest to authors to read and refer to this article which explains in general what amyloid fibrils are: Auriemma Citarella, Alessia, et al. "ENTAIL: yEt aNoTher amyloid fIbrils cLassifier." BMC bioinformatics 23.1 (2022): 1-15.

-In the introduction I suggest putting a short introduction to cardiovascular disease in general and referring to these papers:

De Marco, Fabiola, et al. "Classification of QRS complexes to detect Premature Ventricular Contraction using machine learning techniques." Plos one 17.8 (2022): e0268555.

De Marco, Fabiola, Dewar Finlay, and Raymond R. Bond. "Classification of Premature Ventricular Contraction Using Deep Learning." 2020 Computing in Cardiology. IEEE, 2020.

-At the end of the introduction, I suggest inserting a brief description of the objective of the paper, the dataset used and a very brief description of the results achieved.

- In the introduction, add structure to the paper to make it easier to read.

-Check the references in the method section.

-What are the fields of the dataset? How many people are involved in the study? Clarify and include a better description of the dataset.

-A misunderstanding exists between dataset parameters and model input parameters. Are they identical? Are only some selected? Then why?

-Why are so many more samples taken from people over 50? Does this occur more frequently in people over 50? Explain this point.

-Insert a diagram or table to explain the structure of the dataset well, explaining what they are for each field and category in order to favor the reader.

-What percentage are training, validation and testing chosen?

-Why wasn't the AUC metric used? please if it is possible to insert it.

-The description part of the dataset in the results section should be moved to the dataset subsection.

-It is unclear what descriptions images and tables refer to in the results section. Need to double check and fix. In this way I cannot say that they are clear and adequate

-The conclusion needs improvement. In this way, they cannot be accepted. The authors should provide a summary of their work, report a portion of the findings, and explain how their work has improved the domain.

Reviewer #2: Thank you for the opportunity to review this paper from Laper et colleagues.

I really appreciate the effort to estimate the prevalence of cardiac amyloidosis in a wide national population and I think that machine learning models represent the future of screening for cardiomyopathies and the subject is of great interest, however I have to report my concern about the accuracy of the data presented.

Major revisions:

Labelling data as “reference ATTR-CM” and “not ATTR-CM” was defined by the investigators, potentially introducing a bias, specifically I do not agree with the definitions of “ATTR-CM-like”. From my understanding, in methods you stated that for diagnosis of “ATTR-CM-like” you need 1 cardiac related ICD-10 AND 1 secondary procedure AND 1 mandatory procedure. let me give you an example fulfilling your criteria: patient with a cardiac related ICD 10 AND heart magnetic resonance (mandatory) AND troponin dosage… this example for me it is not suggestive of ATTR-CM unless CMR presents features consistent with amyloid cardiomyopathy, but this is not specified in the text or at least for me it is unclear. I can provide you plenty of combinations that you labelled as “ATTR-CM-like” but they are hardly acceptable.

Similarly with the second possible definition of “ATTR-CM-like” e.g. ICD 10-related wtATTR (organ limited amyloidosis) AND secondary procedure (ECG), it is difficult to accept that this combination is suggestive of “ATTR-CM-like”. For what we know it could be amyloidosis in the carpal tunnel or just in the lung tissue or many other different alternatives but not necessarily “ATTR-CM-like”, isn’t it?

My concern is similar for “hATTR-CM -like”, I provide you another example fulfilling your criteria for “hATTR-CM -like”: Neuropathic heredofamilial amyloidosis (ICD10 criteria) AND skin biopsy (secondary criteria), for me it does not represent a “hATTR-CM-like” but it could simply be a neuropathic amyloidosis with skin biopsy performed and no cardiac involvement.

Secondary procedures criteria may often be just screening procedure? The fact that a patient performed a cardiological screening does not mean that it resulted positive to it.

What I'm trying to prove is that the criteria you have chosen are non specific for amyloid cardiac involvement, or at least are questionable.

I think that we have internationally shared specific criteria for ATTR cardiac involvement, I can accept the fact that in such a large population, referring to ATTR related IC10 can be a valid option however ATTR related ICD-10 are non specific for cardiac involvement and the association with “cardiac related ICD 10” can somehow increase the specificity but the overall result might be unprecise either in “ATTR-CM- like” or in “ATTR-CM-reference” (e.g. neuropathic heredofamilial amyloidosis + aortic valve stenosis fulfill you “ATTR-CM-reference” criteria but for me it can just be a neuropathic amyloidosis with a degree of aortic stenosis in an old patient without amyloidosis proven cardiac involvement)

In conclusion I appreciate the effort but I have to express my concern on most of the definitions you provided and consequently to all the analysis derived from them.

You excluded patients with “blood cancer” to avoid overlap with AL however, all patients with a positive serum or urinary immunofixation rise the suspect of possible AL ( independently from the presence of a blood cancer) and only biopsy proven ATTR should be included, I don’t think this has been done in your study , rising additional biases

Minor revisions:

Is it normal that in the manuscript I repeatedly found “(Supplementary Error! Reference source not found.)”? Maybe there is something wrong in my downloaded manuscript, but it is difficult to understand the refences in the text.

For me it is unclear how you tested the accuracy, sensitivity and specificity of the ML model, from what I understood you classified wtATTR-CM and hATTR-CM based on ICD-10 with criteria not accepted internationally for the diagnosis of cardiac amyloidosis.

6. PLOS authors have the option to publish the peer review history of their article (what does this mean?). If published, this will include your full peer review and any attached files.

Reviewer #1: **Yes: **Fabiola De Marco

Reviewer #2: No

---

## [Author Response · Author response to Decision Letter 0]

26 Sep 2023

We would like to thank the reviewers for the valuable contributions for the article´s improvement. In fact, your insights were key to improve the quality of the article .

Reviewer 1

1. Labelling data as “reference ATTR-CM” and “not ATTR-CM” was defined by the investigators, potentially introducing a bias, specifically I do not agree with the definitions of “ATTR-CM-like”. From my understanding, in methods you stated that for diagnosis of “ATTR-CM-like” you need 1 cardiac related ICD-10 AND 1 secondary procedure AND 1 mandatory procedure. let me give you an example fulfilling your criteria: patient with a cardiac related ICD 10 AND heart magnetic resonance (mandatory) AND troponin dosage… this example for me it is not suggestive of ATTR-CM unless CMR presents features consistent with amyloid cardiomyopathy, but this is not specified in the text or at least for me it is unclear. I can provide you plenty of combinations that you labelled as “ATTR-CM-like” but they are hardly acceptable. 

Response: Thank you for your comment. We have specified in the text clearly the model steps for classification and data source characteristics, as well as the use of an expert panel to mitigate such points mentioned. 

Now the text reflects clearly that those patients were included as potential ATTR-CM cases and then classified with the machine learning model as ATTR-CM-like (lines 95 to 180). This was assessed by the expert panel to identify and propose mitigations on the classification label construction, as you kindly appointed since the dataset did not provide enough clinical information for a deeper assessment. We also stated this better in the study limitations (lines 548 to 569). 

2. Similarly with the second possible definition of “ATTR-CM-like” e.g., ICD 10-related wtATTR (organ limited amyloidosis) AND secondary procedure (ECG), it is difficult to accept that this combination is suggestive of “ATTR-CM-like”. For what we know it could be amyloidosis in the carpal tunnel or just in the lung tissue or many other different alternatives but not necessarily “ATTR-CM-like”, isn’t it?

Response: Thank you for your comment. As mentioned in the answer to your previous comment, we have clarified the methodology for the model label eligibility criteria construction and expert panel validation. We also reflected this in the study limitations.

3. My concern is similar for “hATTR-CM -like”, I provide you another example fulfilling your criteria for “hATTR-CM -like”: Neuropathic heredofamilial amyloidosis (ICD10 criteria) AND skin biopsy (secondary criteria), for me it does not represent a “hATTR-CM-like” but it could simply be a neuropathic amyloidosis with skin biopsy performed and no cardiac involvement.

Response: Thank you for your comment. As mentioned in the answer to your previous comment 1, we have clarified the methodology for the model label eligibility criteria construction and expert panel validation. We also reflected this in the study limitations.

4. Secondary procedures criteria may often be just screening procedure? The fact that a patient performed a cardiological screening does not mean that it resulted positive to it.

Response: Thank you for your comment. We clarified in the methodology that the inclusion criteria were to define the population to have the machine learning model applied (lines 95 to 180). Besides that, it is important to highlight that as this is an administrative claim database study, we have some limitations regarding lab results. Those were also stated in the study limitations text (lines (564 to 569). 

What I trying to prove is that the criteria you have chosen are nonspecific for amyloid cardiac involvement, or at least are questionable.

I think that we have internationally shared specific criteria for ATTR cardiac involvement, I can accept the fact that in such a large population, referring to ATTR related IC10 can be a valid option, however, ATTR related ICD-10 are non specific for cardiac involvement and the association with “cardiac related ICD 10” can somehow increase the specificity but the overall result might be unprecise either in “ATTR-CM- like” or in “ATTR-CM-reference” (e.g. neuropathic heredofamilial amyloidosis + aortic valve stenosis fulfill you “ATTR-CM-reference” criteria but for me it can just be a neuropathic amyloidosis with a degree of aortic stenosis in an old patient without amyloidosis proven cardiac involvement)

Response: Thank you for your comment. We understand the concern raised, but this is an intrinsic limitation of the available information database used for this study. As mentioned before, we tried to mitigate this limitation according to the recommendations of the expert panel feedback. The deeper evaluation your comment proposed would only be possible using a traditional medical chart retrospective assessment, which would be a different study and objective from ours.

In the case of ATTR-CM reference cases, we used the ICD-10 codes, E85.0, E85.1, and E85.2 for hATTR-CM, and E85, E85.3, E85.4, E85.8, and E85.9 combined with cardiac involvement ICD-10 codes. In addition, the same ICD-10 codes were previously published articles, such as Brown et al, 2021 and Jang et al, 2022.

In conclusion, I appreciate the effort but I have to express my concern on most of the definitions you provided and consequently to all the analysis derived from them.

Response: We appreciate all the concerns and questions raised in the review. We considered all of them and included more descriptions of the model construction, datasets, and population included and tested in the methodology. Also, we added more information in the limitations section regarding the absence of lab exam information.

You excluded patients with “blood cancer” to avoid overlap with AL however, all patients with a positive serum or urinary immunofixation rise the suspect of possible AL (independently from the presence of a blood cancer) and only biopsy proven ATTR should be included, I don’t think this has been done in your study, rising additional biases.

Response: We appreciate all the concerns and questions raised by the Reviewer. As mentioned, and explained in the other comments, as this is a claim database study, we have some intrinsic limitations regarding lab results availability. Also, this is also aligned with similar studies using ML performed previously, such as Huda et al, 2021, which is added in the references. 

Is it normal that in the manuscript I repeatedly found “(Supplementary Error! Reference source not found.)”? Maybe there is something wrong in my downloaded manuscript, but it is difficult to understand the refences in the text.

Response: This was a mistake of the reference manager. We have adjusted it in the revised new version. We apologize for that. 

For me it is unclear how you tested the accuracy, sensitivity and specificity of the ML model, from what I understood you classified wtATTR-CM and hATTR-CM based on ICD-10 with criteria not accepted internationally for the diagnosis of cardiac amyloidosis. 

Response: Detailed description included in the methods section and the figure below in the Supplementary material. 

Supplementary Fig 3. Equations for Accuracy, Sensitivity and Specificity calculation

TP: true positive; TN: true negative; FP: false positive; FN: false negative

Reviewer 2

In the introduction I suggest to authors to read and refer to this article which explains in general what amyloid fibrils are: Auriemma Citarella, Alessia, et al. "ENTAIL: yEt aNoTher amyloid fIbrils cLassifier." BMC bioinformatics 23.1 (2022): 1-15.

Response: Thank you for your suggestion. Reference added. 

In the introduction I suggest putting a short introduction to cardiovascular disease in general and referring to these papers: De Marco, Fabiola, et al. "Classification of QRS complexes to detect Premature Ventricular Contraction using machine learning techniques." Plos one 17.8 (2022): e0268555. De Marco, Fabiola, Dewar Finlay, and Raymond R. Bond. "Classification of Premature Ventricular Contraction Using Deep Learning." 2020 Computing in Cardiology. IEEE, 2020.

Response: Thank you. We added a short contextualization of cardiovascular and included the suggested references.

At the end of the introduction, I suggest inserting a brief description of the objective of the paper, the dataset used, and a very brief description of the results achieved.

In the introduction, add structure to the paper to make it easier to read.

Response: Thank you for your suggestion. We adjusted the introduction and added more information on the last part of it. 

Check the references in the method section.

Response: There was an error on the reference manager. We adjusted it for this version. We apologize about it. 

What are the fields of the dataset? How many people are involved in the study? Clarify and include a better description of the dataset.

Response: We included a deeper description of the database in the methodology section. 

A misunderstanding exists between dataset parameters and model input parameters. Are they identical? Are only some selected? Then why?

Response: Some of the dataset parameters were used as model inputs, such as age and procedures performed. 

Why are so many more samples taken from people over 50? Does this occur more frequently in people over 50? Explain this point.

Response: Yes, wtATTR-CM occurs mainly in older populations and this age cutoff was also used in similar studies previously reported. We clarified this point in the methodology section.

Insert a diagram or table to explain the structure of the dataset well, explaining what they are for each field and category in order to favor the reader.

Response: Thank you for your suggestion. We included a specific reference for this, Ali et al, 2019.

What percentage are training, validation and testing chosen?

Response: The training, validation and test steps were performed with 60%, 20% and 20% of patients, respectively. This information was added in the methodology and is also available in the supplementary material figure 2. 

Why wasn't the AUC metric used? please if it is possible to insert it.

Response: The accuracy result generated does not necessarily represent the clinical applicability for the expected result. Despite the universal use in machine learning studies, area under the curve of a receiver operating characteristic curve is not necessarily the best metric to represent clinical applicability and is not easily understandable by many clinicians. Machine learning algorithms will use whatever information are available to achieve the best possible performance in the dataset used. This may include the exploration of unknown confounders that may not be reliable, impairing the algorithm’s ability to generalize to new datasets.

The description part of the dataset in the results section should be moved to the dataset subsection.

Response: We improved the dataset description on the dataset subsection and included results of the methodology applied in the results section

It is unclear what descriptions images and tables refer to in the results section. Need to double check and fix. In this way I cannot say that they are clear and adequate

Response: Thank you for your comment. We have double checked both figures and tables. Also, as the submission system issues the file with figures in separate, usually this makes it harder to evaluate it in the text. 

The conclusion needs improvement. In this way, they cannot be accepted. The authors should provide a summary of their work, report a portion of the findings, and explain how their work has improved the domain.

Response: Information added in the conclusion section. 

“In our study, we were able to characterize this population demographically, clinically (considering their ICD-10 codes and procedures performed), and to identify the HCRU related to ATTR-CM management. The use of ML as a tool to identify potential patient of underdiagnosed diseases can be a hallmark for public health resources allocation and medical education strategies. In addition, our findings may be useful to support the development of health guidelines and policies to improve diagnosis, treatment and to cover unmet medical needs of patients with ATTR-CM in Brazil.”

REFERENCES 

• Brown D, Vera-Llonch M, Perez J, et al. USE OF COMMERCIAL CLAIMS DATA TO ESTIMATE TRANSTHYRETIN-AMYLOID CARDIOMYOPATHY PREVALENCE AND INCIDENCE IN THE US. J Am Coll Cardiol. 2021 May, 77 (18_Supplement_1) 884. https://doi.org/10.1016/S0735-1097(21)02243-9 

• Huda, A., Castaño, A., Niyogi, A. et al. A machine learning model for identifying patients at risk for wild-type transthyretin amyloid cardiomyopathy. Nat Commun 12, 2725 (2021). https://doi.org/10.1038/s41467-021-22876-9

• Jang SC, Nam JH, Lee SA, An D, Kim HL, Kwon SH, Lee EK. Clinical manifestation, economic burden, and mortality in patients with transthyretin cardiac amyloidosis. Orphanet J Rare Dis. 2022 Jul 15;17(1):262. doi: 10.1186/s13023-022-02425-3. PMID: 35840997; PMCID: PMC9287852.

---

## [Decision Letter · Decision Letter 1]

9 Nov 2023

PONE-D-22-32110R1Assessment of potential transthyretin amyloid cardiomyopathy cases in the Brazilian public health system using a Machine Learning ModelPLOS ONE

Dear Dr. Silva Julian,

Thank you for submitting your manuscript to PLOS ONE. After careful consideration, we feel that it has merit but does not fully meet PLOS ONE’s publication criteria as it currently stands. Therefore, we invite you to submit a revised version of the manuscript that addresses the points raised during the review process.

We look forward to receiving your revised manuscript.

Kind regards,

Giuseppe Limongelli

Academic Editor

PLOS ONE

Journal Requirements:

Additional Editor Comments:

Dear author

Thank you for re-submitting your manuscript and for addressing the points raised by the reviewers. We would suggest a few minor changes:

- review punctuation (spaces after brakets)

- review the reference in line 309

- add a figure to help understand the composition of the dataset visually

looking forward to receive the manuscript

Reviewers' comments:

Reviewer's Responses to Questions

**Comments to the Author**

1. If the authors have adequately addressed your comments raised in a previous round of review and you feel that this manuscript is now acceptable for publication, you may indicate that here to bypass the “Comments to the Author” section, enter your conflict of interest statement in the “Confidential to Editor” section, and submit your "Accept" recommendation.

Reviewer #1: All comments have been addressed

2. Is the manuscript technically sound, and do the data support the conclusions?

Reviewer #1: Yes

3. Has the statistical analysis been performed appropriately and rigorously? 

Reviewer #1: Yes

4. Have the authors made all data underlying the findings in their manuscript fully available?

Reviewer #1: Yes

5. Is the manuscript presented in an intelligible fashion and written in standard English?

Reviewer #1: Yes

6. Review Comments to the Author

Reviewer #1: This work present a retrospective study aimed to identify potential transthyretin cardiac amyloidosis (ATTR-CM) cases in the Brazilian public health system using predictive machine learning to improve diagnosis and treatment.

All the comments have been addressed.

Minor comments:

- - Please, pay attention to spaces in some part of the text and before some brackets;

- Line 309: there is an error for the reference source;

- Some figures could be improved.

7. PLOS authors have the option to publish the peer review history of their article (what does this mean?). If published, this will include your full peer review and any attached files.

Reviewer #1: No

---

## [Author Response · Author response to Decision Letter 1]

11 Nov 2023

1. Review punctuation (spaces after brackets).

Response: Thank you for your suggestion. We reviewed all the spaces after brackets.

2. Review the reference in line 309.

Response: Thank you! We reviewed all the references in the article.

3. Add a figure to help understand the composition of the dataset visually.

Response: Thank you for your suggestion, we included one figure as supplementary figure 2.

---

## [Editor Report · Decision Letter 2]

15 Nov 2023

Assessment of potential transthyretin amyloid cardiomyopathy cases in the Brazilian public health system using a Machine Learning Model

PONE-D-22-32110R2

Dear Dr. Silva Julian,

We’re pleased to inform you that your manuscript has been judged scientifically suitable for publication and will be formally accepted for publication once it meets all outstanding technical requirements.

Kind regards,

Giuseppe Limongelli

Academic Editor

PLOS ONE
---

## [Editor Report · Acceptance letter]

28 Nov 2023

PONE-D-22-32110R2 

Assessment of potential transthyretin amyloid cardiomyopathy cases in the Brazilian public health system using a Machine Learning Model 

Dear Dr. Silva Julian:

I'm pleased to inform you that your manuscript has been deemed suitable for publication in PLOS ONE. Congratulations! Your manuscript is now with our production department. 

Kind regards, 

on behalf of

Dr. Giuseppe Limongelli 

Academic Editor

PLOS ONE